# Synchrotron Radiation Study of Gain, Noise, and Collection Efficiency of GaAs SAM-APDs with Staircase Structure

**DOI:** 10.3390/s22124598

**Published:** 2022-06-17

**Authors:** Matija Colja, Marco Cautero, Ralf Hendrik Menk, Pierpaolo Palestri, Alessandra Gianoncelli, Matias Antonelli, Giorgio Biasiol, Simone Dal Zilio, Tereza Steinhartova, Camilla Nichetti, Fulvia Arfelli, Dario De Angelis, Francesco Driussi, Valentina Bonanni, Alessandro Pilotto, Gianluca Gariani, Sergio Carrato, Giuseppe Cautero

**Affiliations:** 1Department of Engineering and Architecture, University of Trieste, 34127 Trieste, Italy; marco.cautero@phd.units.it (M.C.); carrato@units.it (S.C.); 2Elettra-Sincrotrone Trieste S.C.p.A., Area Science Park Basovizza, 34149 Trieste, Italy; ralf.menk@elettra.eu (R.H.M.); alessandra.gianoncelli@elettra.eu (A.G.); dario.deangelis@elettra.eu (D.D.A.); valentina.bonanni@elettra.eu (V.B.); gianluca.gariani@elettra.eu (G.G.); giuseppe.cautero@elettra.eu (G.C.); 3Istituto Nazionale di Fisica Nucleare Sezione di Trieste, 34127 Trieste, Italy; matias.antonelli@ts.infn.it (M.A.); fulvia.arfelli@ts.infn.it (F.A.); 4Department of Medical Imaging, University of Saskatchewan, Saskatoon, SK S7N 5A2, Canada; 5Polytechnic Department of Engineering and Architecture, University of Udine, 33100 Udine, Italy; pierpaolo.palestri@uniud.it (P.P.); francesco.driussi@uniud.it (F.D.); pilotto.alessandro@spes.uniud.it (A.P.); 6Laboratorio TASC, Area Science Park Basovizza, Consiglio Nazionale delle Ricerche-Istituto Officina dei Materiali, 34149 Trieste, Italy; biasiol@iom.cnr.it (G.B.); Dalzilio@iom.cnr.it (S.D.Z.); einsteinhartova@gmail.com (T.S.); 7Department of Physics, University of Trieste, 34127 Trieste, Italy; camilla.nichetti@elettra.eu (C.N.)

**Keywords:** X-ray photodetector, GaAs separate absorption multiplication avalanche photodiode (GaAs SAM-APD), synchrotron radiation, collection efficiency, staircase structure

## Abstract

In hard X-ray applications that require high detection efficiency and short response times, such as synchrotron radiation-based Mössbauer absorption spectroscopy and time-resolved fluorescence or photon beam position monitoring, III–V-compound semiconductors, and dedicated alloys offer some advantages over the Si-based technologies traditionally used in solid-state photodetectors. Amongst them, gallium arsenide (GaAs) is one of the most valuable materials thanks to its unique characteristics. At the same time, implementing charge-multiplication mechanisms within the sensor may become of critical importance in cases where the photogenerated signal needs an intrinsic amplification before being acquired by the front-end electronics, such as in the case of a very weak photon flux or when single-photon detection is required. Some GaAs-based avalanche photodiodes (APDs) were grown by a molecular beam epitaxy to fulfill these needs; by means of band gap engineering, we realised devices with separate absorption and multiplication region(s) (SAM), the latter featuring a so-called staircase structure to reduce the multiplication noise. This work reports on the experimental characterisations of gain, noise, and charge collection efficiencies of three series of GaAs APDs featuring different thicknesses of the absorption regions. These devices have been developed to investigate the role of such thicknesses and the presence of traps or defects at the metal–semiconductor interfaces responsible for charge loss, in order to lay the groundwork for the future development of very thick GaAs devices (thicker than 100 μm) for hard X-rays. Several measurements were carried out on such devices with both lasers and synchrotron light sources, inducing photon absorption with X-ray microbeams at variable and controlled depths. In this way, we verified both the role of the thickness of the absorption region in the collection efficiency and the possibility of using the APDs without reaching the punch-through voltage, thus preventing the noise induced by charge multiplication in the absorption region. These devices, with thicknesses suitable for soft X-ray detection, have also shown good characteristics in terms of internal amplification and reduction of multiplication noise, in line with numerical simulations.

## 1. Introduction

Single-photon detectors represent the ultimate sensitivity limits for any quantum radiation sensor and are employed in different science fields, such as medical imaging, hazard, and threat detection [1,2], (bio)-photonics [3], high-energy physics, and astrophysics [4,5]. For photons in the visible range, the best-known and most widely used solid-state devices are silicon (Si) APDs. As will be discussed in detail in Section 2, APDs are extremely efficient and sensitive due to their ability to exploit the avalanche multiplication of the photogenerated carriers. By amplifying the photo-current above the noise floor of the read-out circuit, they can dramatically improve the signal-to-noise ratio (SNR). This property makes them very effective in sensing extremely weak signals.

The most successful Si APDs [6] and the more recently introduced Si photomultipliers [7] have limited sensitivity caused by their (indirect) band gap of about 1.12 eV at 300 K. Si-based single-photon detectors can indeed achieve direct light detection with fast time responses and low jitters (150 ps) only by thin n-epitaxial layers (2–4 μm), which limit their application to photons in the spectral range from 350 to 800 nm. For shorter wavelengths (<1 nm), i.e., in the X-ray region, their absorption capabilities decrease rapidly with increasing photon energy. This limits their use in many synchrotron radiation and free-electron laser experiments, since X-ray absorption scales with a high power of the atomic number, at least in the energy range dominated by the photoelectric effect.

In the literature, there are no reports of silicon-based APDs capable of detecting hard X-rays directly and with high efficiency. In order to increase efficiency, keeping an acceptable thickness, solutions were proposed based either on the stacking and/or tilting of several APDs [8], or on the conversion of the X-ray photons to visible light by high-Z scintillators, such as Pb-doped plastics [9].

Compared to silicon devices, compound semiconductors based on III–V elements, such as GaAs, have some unique properties, such as a higher density (5.32 g/cm${}^{3}$ vs. 2.33 g/cm${}^{3}$ of Si), a high effective atomic number (32 vs. 14 for Si [10]), a direct energy band gap, high electric-breakdown fields (4 × 10${}^{5}$ V/cm vs. 3 × 10${}^{5}$ V/cm for Si), and a high electron mobility (8000 cm${}^{2}$/Vs vs. 1350 cm${}^{2}$/Vs for Si), which makes them suitable for photonic, radiofrequency, and high-power device technologies [11]. The absorption lengths for medium and hard X-ray energies are substantially shorter in GaAs than in Si, owing to the higher effective atomic number. This results in a much higher absorption coefficient and higher detection efficiency when compared to Si sensors of the same thickness or, conversely, having a shorter absorption length allows to have thinner devices; that, combined with the larger electron mobility of GaAs, translates into shorter response times, making GaAs sensors particularly suitable for time-resolved experiments. Their applications include the detection of scintillator luminescent light as an alternative to Si and Ge [12], GaAs-based electronic and optoelectronic devices [13] for LiDAR and ranging applications [14], and the development of efficient photovoltaic devices [15]. Furthermore, in recent years, sensors based on chromium-compensated GaAs have shown to be very promising for X-ray and γ-ray spectroscopic imaging [16,17,18,19].

Thanks to the characteristics listed above, in recent years, GaAs [20], as well as wide-band gap materials, such as CVD diamond [21,22] and SiC [23] or Al${}_{x}$Ga${}_{1-x}$ As [24], have been studied as Si alternatives for the production of X-ray detectors. In particular, there is growing interest in high-energy fluorescence spectroscopy with GaAs-based detectors [25], not only for the aforementioned higher efficiency compared to silicon but also for the possibility of working at room temperature, as opposed to the detectors currently used for high energies, based on germanium, which require liquid nitrogen cooling.

With GaAs-based APDs, there are two main factors responsible for an increase in multiplication noise: the possibility of the multiplicative process starting in random areas of the device and the occurrence of such a process for both electrons and holes with similar probabilities. For these reasons, the new state-of-the-art devices are characterised by two important properties: on the one hand, a separation is made between the photon–electron conversion zone (absorption layer, ideally immune from multiplication events) and the multiplication layer; on the other hand, the use of band gap engineering is exploited to create multiplication layers based on the superlattice: they result in a staircase profile of the conduction band, which promotes only electron impact ionisations at discrete locations, increasing the differences between the effective ionisation coefficients of holes and electrons. Devices fabricated with these features are then called separate absorption and multiplication APDs (SAM-APDs) [26,27].

In this type of device, the thickness of the absorption region is a critical parameter that ultimately defines the maximum energy that is detectable with a certain confidence level. If photon absorption takes place in the multiplication region, this causes a substantial increase in the multiplication noise. Furthermore, a critical aspect that could degrade the SNR is that multiplication may also occur in the absorption region.

The GaAs SAM-APDs presented in this work were expressly developed to lay the groundwork for the subsequent development of hard X-ray detectors. For that reason, these devices feature shallow absorption regions, which make them less efficient for energies higher than a few keV; nevertheless, they are suitable for investigating some important phenomena that are critical as the device becomes thicker. In particular, two interesting aspects emerged from the results shown here: first, the capability of these devices to operate and achieve good collection efficiencies without the depletion of the absorption region (i.e., before punch-through takes place), regardless of the thickness of the absorption region itself. Second, the onset of the desired multiplication mechanism for voltages below the punch-through threshold.

The results of the measurements carried out with both conventional and synchrotron light sources are reported and discussed in comparison to simulations performed alongside.

## 2. Materials and Methods

### 2.1. Mode of Operation

For an X-ray photon with energy $E_{\gamma }$, where the photoelectric absorption is the most probable interaction with the matter (being a species with atomic number *Z*), the mass absorption coefficient μ is proportional to [28]
(1)$$\mu \left(E_{\gamma },Z\right)\propto \frac{Z^{5}}{E_{\gamma }^{3.5}}.$$

The X-ray mean-free path λ within the detector material, i.e., the depth at which the initial X-ray intensity decreases by 1/e, is given by λ=1/μ. Owing to the difference in the atomic number, the attenuation length of GaAs at hard-X-ray energies is considerably shorter than that of Si. For example, at 20 keV, the Si attenuation length ($\lambda_{Si}=1038\;$μm) is about 20 times shorter than the GaAs attenuation length ($\lambda_{GaAs}=44\;\mu$m).

For monoenergetic hard X-rays, the quantum efficiency (QE) is given by
(2)$$QE=e^{-\mu_{w}\;\cdot \;\Delta x_{w}}\;\cdot \;\left(1-e^{-\mu_{GaAs}\;\cdot \;\Delta x_{GaAs}}\right),$$
where $\mu_{w}$ and $\mu_{GaAs}$ are the effective absorption coefficients of the window and detector materials, respectively, and $\Delta x_{w}$ and $\Delta x_{GaAs}$ are their thicknesses [29]. The first factor in Equation (Equation 2) represents the transmission through the entrance window (which can be neglected for sufficiently high X-ray energies) and the second factor (in brackets) is the absorption in the active part of the detector. For instance, at 20 keV (≈0.0620 nm), the absorption in a 100- μm-thick GaAs detector is greater than 90%.

By using Equation (Equation 2), the photogenerated current in the GaAs APD for an incident photon flux $\Phi_{0}$ (in units [1/s]) of monoenergetic X-rays with energy $E_{\gamma }$ can be expressed as
(3)$$I\left(E_{\gamma }\right)=\Phi_{0}\;\cdot \;\frac{E_{\gamma }}{E_{e-h}}\;\cdot \;q\;\cdot \;QE\;\cdot \;\epsilon \;\cdot \;M,$$
where *q* is the electron charge, $E_{e-h}=4.2\;$eV is the average energy required to create an electron/hole pair in GaAs [30], ϵ denotes the charge collection efficiency, and *M* is the internal avalanche multiplication of the photogenerated carriers in the APD (Equations (Equation 2) and (Equation 3) hold only if photons do not generate e-h pairs in the multiplication layer/substrate). This multiplication is exploited to detect very weak signals and to improve the SNR. The multiplication relies on impact ionisation, which occurs when the kinetic energy of photoinduced charge carriers, (e.g., electrons) is greater than the $E_{e-h}$. This process is an advantage over the non-multiplicative charge collection as long as the noise induced by the multiplication, represented through the excess noise factor (ENF), is kept low (ideally close to unity) by properly choosing materials and structures. Otherwise, given a specific input-referred noise of the front-end electronics, a higher input signal obtained through charge multiplication can still result in a signal-to-noise ratio lower than what is achievable with less or no internal multiplication.

Excess multiplication noise results from the stochastic nature of the impact ionisation process that amplifies the photogenerated current [31]. The ENF is defined as the ratio between the actual noise of the APD current and the noise of a device with noiseless multiplication (shot noise only). Let *n* be the number of multiplied output carriers originating from *a* primary carriers generated by a Poisson process, then the ENF can be expressed as [32]
(4)$$ENF=\frac{\mathrm{var}(n)}{M^{2}\;\cdot \;\mathrm{var}\left(a\right)}.$$

The ENF in APDs is particularly high when electron and hole ionisation coefficients, α and β, respectively, are very similar, which is exactly the case for GaAs [33]. For this reason, by taking advantage of band gap engineering, we developed APD devices featuring multiplication layers based on a GaAs/AlGaAs superlattice staircase structure (Figure 1). Such a structure promotes electron impact ionisation at specific locations and subsequently increases the difference between the effective ionisation coefficients of holes and electrons, dramatically reducing the ENF [32].

Moreover, as discussed previously, the location at which the e-h pairs are photogenerated is fundamental for noise reduction. Firstly, the photogeneration should not take place inside the multiplication region [30], as that would result in each carrier encountering a randomly different number of multiplication stages, depending on its original location. The presence of a separate absorption region of a specific thickness ensures that most of the photons (below a given energy) are absorbed within such a layer and, thus, they do not generate carriers in the multiplication region. Secondly, for similar reasons, multiplication should not occur in the absorption region, otherwise, photogenerated carriers would be randomly multiplied before entering the multiplication region. The introduction of a separation layer between the two regions, referred to as the p-doped layer, can prevent the depletion of the absorption region while the rest of the device is adequately biased; in this way, the field is negligible in the absorption region and, consequently, its contribution to multiplication is minimised.

Thanks to this layer, by applying a reverse voltage, initially the electric field increases in the multiplication region only. In this phase, the photocurrent is very low, close to the value of the dark current, as the photogenerated electrons are not able to pass through the potential barrier introduced by the p-doped layer itself. As the bias increases, this barrier is reduced, and once the barrier is completely removed, the electrons can now move from the absorption region to the multiplication region, increasing the photocurrent. The bias at which the barrier disappears is called the punch-through voltage. By further increasing the reverse bias, the absorption region becomes depleted and its electric field increases, forcing the electrons to drift towards the multiplication layer. This is a typical mode of operation of SAM-APDs [14].

A possible disadvantage of this mode of operation is that the multiplicative process could also take place in the absorption region if the electric field is too high and this increases the multiplication noise. To avoid such a drawback, the SAM-APDs presented in this manuscript work differently with respect to classic devices: they were fabricated by depositing a δ p-doped sub-monolayer of C atoms with a sufficiently high acceptor concentration to avoid punch-through [34] but, at the same time, they were thin enough to allow most of the electrons produced in the absorption region enter the multiplication region. The advantage of this approach is that the electric field is applied in the multiplication region only, provided that the current can flow despite the potential barrier introduced by the δ layer.

In the past, in-depth studies have been carried out on the behaviours of these devices regarding the doping levels of the various layers, the number of multiplication steps, and the fundamental role of the δ p-doped layer [35,36,37,38]. Thanks to these studies, it was found that a δ p-doping layer of carbon atoms of 2.5 × 10${}^{12}$ cm${}^{-2}$ is necessary to keep the absorption region unbiased over the whole range of reverse biases, up to the breakdown voltage.

### 2.2. Device Growth and Fabrication

For the study at hand, several GaAs APDs have been grown by the molecular beam epitaxy on a 500-μm-thick heavily Si-doped (2 × 10${}^{18}$ cm${}^{-3}$) n-type GaAs (001) substrate, following the procedure outlined in [35]. The resulting layered structure is reported in Figure 1. Growth temperature was set at 580 ${}^{\circ }$C for all the layers, with Ga and As ${}_{4}$ partial pressures of about 7 × 10${}^{-7}$ and 1 × 10${}^{-5}$ Torr, respectively, corresponding to a GaAs growth rate of 1 μm/h. A 1-μm-thick intrinsic multiplication layer was grown first. This layer included a staircase structure with 12 stages (repetitions). Each stage consisted of 35 nm of GaAs, 25 nm of Al${}_{0.45}$Ga${}_{0.55}$As, and 20 nm of a linearly graded region formed by a digital alloy where the Al content was reduced from 45% to 1%.

Above the staircase structure, a 35-nm-thick GaAs spacer was grown, followed by a δ p-doped sub-monolayer of carbon atoms. As pointed out before, this layer controls the electrical separation between the absorption and multiplication regions, ensuring that the applied voltage drops mainly in the multiplication region. Above the δ layer, three different thicknesses ($d_{abs}$ = 0.3 μm, 4.5 μm, or 15 μm) of intrinsic GaAs were deposited, determining three different types of devices. Finally, the samples were capped with a 200-nm highly p-doped (5 × 10${}^{18}$ cm${}^{-3}$) GaAs contact layer. A Cr/Au (chromium 10 nm and gold 50 nm) layer was then deposited as an ohmic contact to connect the preamplifiers via wire bonding. In order to examine whether the presence of any traps in the metal–semiconductor interface played a significant role in lowering efficiency, a problem already reported in the literature [39], this contact layer covered only a portion of the entry window (more precisely, a thin half-moon-shaped area was spared during the deposition), so that photons could impinge either onto the gold contact or directly onto the GaAs. The rear contact, instead, was composed of 50 nm of AuGe, 10 nm of Ni, and 40 nm of Au. Mesa diodes with 200-μm diameters were chemically etched by the solution H${}_{3}$PO${}_{4}$: H${}_{2}$O${}_{2}$:H${}_{2}$O (3:1:50). Furthermore, to reduce the leakage currents, the devices were passivated with Al${}_{2}$O${}_{3}$, grown by sputtering techniques.

### 2.3. Device Simulations

Simulations were performed by using the Sentaurus TCAD software suite [40] and were mainly used to reproduce the experimental capacitance versus bias, as well as the dark current under different bias conditions. A vertical cross-section of the device was modelled as a one-dimensional structure, featuring the doping levels and depths of the various regions described in the previous section. Particular care was given to the multiplication region. To replicate the graded Al${}_{x}$Ga${}_{1-x}$ As section, a digital alloy pattern was used, progressively increasing the width of Al${}_{0.45}$Ga${}_{0.55}$As layers that were intercalated with GaAs layers.

The device was simulated at a temperature of 300 K and general physics included Fermi statistics, a mobility model that considers the high-field saturation as well as the doping dependence, and the Shockley–Read–Hall (SRH) model, which includes both doping and electric-field dependence. “HeteroInterface” and “thermionic” models [41] were employed to handle the transport at the interfaces between GaAs and Al${}_{0.45}$Ga${}_{0.55}$As layers. A 20 nm global mesh was used on the whole design and a much finer 0.001 nm was applied to the interface regions between different compounds or in regions with doping variations, especially around the δ layer, to better simulate the device behaviour in the presence of steep changes of the band profile. In particular, the δ layer was modelled as a thin Gaussian charge distribution with FWHM of 0.01 nm because of technical limitations in the modelling of the sub-monolayer. For the same reason, abrupt doping variations were smoothed out using Gaussian profiles.

### 2.4. Measurements

#### 2.4.1. Capacitance and I–V Curves

Capacitance measurements under increasing reverse biases were performed by using a high-precision LCR meter (HP4284A, Keysight Technology, Santa Rosa, CA, USA) at 1 MHz, providing a 0.05% basic accuracy. In addition, a custom-made acquisition system was employed to measure the current versus reverse bias characteristics (I–V curves). The latter is based on a variable-range (±2.5 nA, ±2.5 μA, ±2.5 mA with 250 fA${}_{rms}$, 25 pA${}_{rms}$, 25 nA${}_{rms}$ error, respectively) trans-impedance amplifier (TIA) characterised by a bandwidth of 4.8 kHz with coarse ranges and 1.0 kHz with the fine range. The delta-sigma analogue-to-digital converter (ADS 1252, Texas Instruments, Dallas, TX, USA) sampled the output voltage of the input TIA stage at 26 kHz and performed digital low-pass filtering with a 5.6-kHz bandwidth. The acquisition software further averaged these values; thus, the acquired data can be considered as DC currents.

#### 2.4.2. ENF

The term var(n) in Equation (Equation 4) represents the variance of the multiplied output carriers, i.e., the total output current noise. Considering a bandwidth *B*, this term can be obtained by integrating the current spectral density $S_{i}$ in the bandwidth *B*. The term var(a), instead, represents the variance of the photogenerated carriers without multiplication. Considering the Poisson process, Equation (Equation 4) can be operatively rewritten as
(5)$$ENF=\frac{S_{i}\;\cdot \;B}{M^{2}\;\cdot \;2qI_{ph}\;\cdot \;B}$$
where $I_{ph}$ is the DC value of the photogenerated current inside the absorption layer before the multiplication process.

In order to measure $S_{i}$, a second TIA was developed, which has a trans-resistance of 5.6 kΩ and a cutoff frequency of 11 MHz. Its output voltage was fed into a signal analyser (Agilent EXA N9010A) through a decoupling capacitor.

#### 2.4.3. Laser Measurements

It was useful to perform the first characterisation by using a visible laser light (although the GaAs APDs are mainly designed for higher photon energies). The relatively low photon energy (2.33 eV) ensured a single electron-hole pair production for each photon and avoided photon penetration inside the multiplication region, even for devices with the thinnest absorption region (see Figure 2a). Such conditions allowed ENF estimation using Equation (Equation 4) without superposition of effects stemming from multiple photogenerated pairs and photogeneration in the multiplication region.

#### 2.4.4. Synchrotron Radiation Measurements

Laser measurements allowed us to obtain some preliminary results. However, they had limited quantitative significance, as the focused laser spot was slightly larger than the entrance window of the APDs and, consequently, the absolute numbers of impinging photons were known with limited precision. Moreover, it would be beneficial to generate charge carriers at different absorption depths, thus monitoring if the overall efficiency (quantum efficiency/charge collection efficiency) changed significantly with the charge generation spatial coordinates; in particular, with the distance from the multiplication layer.

To overcome these limitations, quantitative measurements were carried out with a low-energy synchrotron radiation beamline at Elettra Sincrotrone Trieste. The beamline (TwinMic [44]) has a short undulator and focusing optics, generating a sub-micrometric monochromatic pencil beam in the energy range between 400 and 2200 eV, a range in which the GaAs attenuation length varies significantly (see Figure 2b). The beamline is equipped with scanning stages and eight spectroscopic Silicon drift detectors, which allow performing two-dimensional mesh scans with sub-micrometric pitches while acquiring the X-ray fluorescence spectra of the sample when the monochromatised X-ray beam is impinging perpendicularly. In this way, it is possible to carry out APD surface chemical analyses while simultaneously acquiring maps of the photogenerated current. Typical step sizes for the experiment discussed here were 10 μm × 10 μm, while the beam size was set to 2 μm, providing a flux of 2 × 10${}^{10}$ photons/s. The incident photon flux was monitored with an AXUV 100 PIN diode, which was previously calibrated by the Physikalisch-Technische Bundesanstalt (Berlin, Germany). For the considered photon energy range, the spectral responsivity was 0.2705 ± 4 × 10${}^{-3}$ A/W. For the study at hand, five distinct photon energies were chosen (940, 1090, 1500, 1705, and 2010 eV) in order to change the distribution of the e-h generation, which depended on the associated attenuation lengths (respectively, 860 nm, 1.2 μm, 360 nm, 450 nm, and 680 nm, as shown in Figure 2b). As expected, the depth-dependent transmission in Figure 3 shows that a higher percentage of photons were absorbed deeper inside the absorption region as the attenuation length increased. Therefore, if defects in the absorption region were responsible for a reduction of the collection efficiency, the latter would also change with the attenuation length.

## 3. Results

### 3.1. Capacitance and I–V Curves

In Figure 4a, we show the capacitance measurements and the simulations as functions of the reverse bias. At low biases, the first decrease in capacitance can be observed for all the curves, corresponding to the depletion of the multiplication layer, which then remains almost constant up to the breakdown voltage (37 V). The obtained values are in good agreement with the theoretical capacitance for a parallel-plate capacitor (see the horizontal dashed line in Figure 4a) featuring an area of A=π·(100 μm)${}^{2}$ and separation of d=1 μm filled with GaAs ($C_{th}=\epsilon_{0}\epsilon_{r}A/d=3.59\;$pF, where $\epsilon_{0}=$8.85 × 10${}^{-14}$ F/cm is the vacuum permittivity and $\epsilon_{r}=12.9$ is the GaAs relative permittivity), where *d* is the thickness of the multiplication region. The offsets between the measurements of the three types of devices are due to small geometrical differences in the areas, caused by the anisotropic nature of the etching process. In fact, differences of 0.1 pF can usually also be seen in C-V curves of different devices of the same batch.

On the other hand, the different slopes of the simulated curve can be attributed to an approximated reproduction of the doping profiles in the presence of abrupt doping changes, such as in the δ p-doped layer, which could translate into a slightly wider or a shorter depleted region at a certain bias.

Differently from what happens in devices showing punch-through, no further capacitance decrease is measured at higher voltages. For the devices analysed in this work, the punch-through is never reached, thus the capacitance only depends on the thickness of the multiplication region, which, in our case, is (nominally) the same for all tested devices (hence a single simulated curve). Moreover, since the simulations do not include impact ionisation (and, thus, breakdown), it is possible to see that, beyond the breakdown voltage, when the punch-through takes place, the capacitance begins to decrease as expected.

Figure 4b shows that the simulated dark current of the device modelled with the digital alloy (blue curve) presents an overshoot with respect to the measurement for voltages below 15 V. This strange behaviour, never seen in the experimental devices, could be attributed to a wrong response of the SRH model when coupled with the heterointerfaces of the digital-alloy structure. In particular, this layer is very thin; effects related to tunnelling, not included in the model, may be present. For example, it was observed that when replacing the digital alloy with a graded transition (where the molar fraction of the Al was linearly increased up to 0.45), the hump disappeared, but the current also changed by orders of magnitude. On the other hand, if only GaAs was considered, the current increased and the current step caused by the punch-through became wider. Clearly, the presence of many thin layers with different compositions pushes the drift-diffusion model with local SRH to its limit. Obtaining a better modelling description of the dark current is however beyond the aim of the current paper.

A typical dark current I–V curve is depicted as a blue solid line in Figure 5a. An exponential increase up to the breakdown voltage can be observed; moreover, it is evident that, up to that voltage, there is no appreciable further increase in the current due to multiplication. This is consistent with numerical simulations [45], which show that in a correctly manufactured device the dark current comes essentially from generation/recombination in the multiplication region, and, therefore, it does not undergo the entire multiplicative process.

The I–V curves of the three types of devices, which were irradiated with the same laser power ($P_{laser}\approx$ 50 μW, λ=532nm), are shown in Figure 5a. For all devices, three distinct regions can be distinguished: firstly, in the reverse-bias range from about 5 to 25 V, an exponential (therefore linear on a semi-logarithmic scale) current increase is observable. This exponential trend is consistent with a progressive lowering of the potential barrier of the δ layer and is hardly attributable to multiplication since the electric field in the multiplication region is still too low. Then, from 25 to about 37 V, a further increase due to multiplication is visible, and eventually breakdown occurs for bias voltages above 37 V. From Figure 5a, it is also clear that the current values in all three device types are comparable (when exposed to the same photon flux), even if the absorption region length varies by a factor of 50; therefore, there is limited loss during electron travelling through the absorption region.

In Figure 5b, we report a comparison between “normalised” dark and photoinduced currents, in order to highlight the fact that the multiplication starting at 25 V does not involve the dark current. As will be discussed in Section 3.2, normalisation should be understood as the division of the measured currents by the exponential function obtained by interpolating the data in the range 7.5–22.5 V.

### 3.2. Excess Noise Factor

As explained before, the ENF determination requires the precise measurement of the APD power spectral density $S_{i}$ (Equation (Equation 5)). The front-end input-referred noise floor is shown in Figure 6a (blue trace), which has a minimum value of 3 × 10${}^{-24}$ A${}^{2}$/Hz at 1 MHz and which is almost entirely caused by thermal noise of the feedback resistor ($i_{n}^{2}=4kT/R_{f}$ = 2.9 × 10${}^{-24}$ A${}^{2}$/Hz) at room temperature. For higher frequencies, the noise floor increases owing to the input-referred voltage noise of the operational amplifier ($e_{n}$ = 4.3 nV/$\sqrt{\mathrm{Hz}}$ [46]), which, above the cut-off frequency $f_{c}=2\pi R_{f}C_{f}=11$ MHz, is amplified by a factor $C_{in}/C_{f}$. The total input capacitance (device and wires) is approximately 25 pF; therefore, the resulting maximum noise floor (expressed as the power spectral density of the current) is approximately ${\left(e_{n}\;\cdot \;C_{in}\right)}^{2}/{\left(R_{f}\;\cdot \;C_{f}\right)}^{2}=5.9$ × 10${}^{-23}$ A${}^{2}$/Hz.

The red trace in Figure 6a represents a typical spectrum in the presence of light: it is almost flat, as expected for shot noise. At low frequencies, a typical 1/f noise is visible. Therefore, we considered the flat bandwidth region from 1.5 to 2 MHz to estimate the $S_{i}$ required in Equation (Equation 5).

From Equation (Equation 5), it is evident that the ENF characterisation requires accurate knowledge of the gain *M*. Typically, in traditional APD devices, the gain is estimated by normalising the current at voltages just above the punch-through voltage since the photo-current is relatively flat and there is no multiplication yet. However, if the multiplication process begins before the punch-through voltage, this leads to an inaccurate gain estimation [48].

The condition of the study at hand is even more complex since punch-through is never achieved; therefore, in order to estimate the value of the gain *M*, the trend of the I–V curve on a semi-logarithmic scale was interpolated with a straight line from 7.5 to 22.5 V, where there is no evident deviation from the exponential trend. Such interpolating line represents the exponential growth of the current with no multiplicative effect; therefore, the gain was calculated as the ratio between the measured current and such an exponential trend, namely
(6)$$M\left(V\right)=\frac{I_{m}^{ph}\left(V\right)-I_{m}^{dark}\left(V\right)}{a\;\cdot \;e^{b\;\cdot \;V}},$$
where *a* and *b* are the parameters of the exponential trend extracted through the interpolation. The gains obtained with each type of device are reported in Figure 6b.

It is common to compare the results with the theoretical trend of the local model: the ENF may be written in terms of both the gain *M* and the ratio of the ionisation coefficients for electrons and holes, namely k=α/β. This relation can be expressed in the well-known form [47]
(7)$$ENF\left(M\right)=k\;\cdot \;M+\left(1-k\right)\left(2-\frac{1}{M}\right).$$

For each device type, the $S_{i}$ was measured as the reverse bias voltage increased and the resulting ENFs are reported in Figure 6c, where a value of *k* ranging from 0.25 to 0.35 is calculated for different devices. Hence, the lower value of *k*, compared with the one found in simple GaAs PIN diodes, which approximately equals 1 [49], clearly indicates that the staircase structure used for the multiplication region is effectively hindering the contribution of holes in the avalanche process. This implies, as evident from Equation (Equation 7), a reduction of the ENF.

The measured values for *k*, *M*, and the dark current show that the developed devices are excellent photon detectors working in the linear regime in spectral regions that could range, depending on the thickness of the absorption region, from a few eV up to some keV. In the literature, for similar devices based on GaAs–AlGaAs, the reported values of *k* range from 0.15 to 0.4 [50,51,52], obtained either by developing the staircase structure for the multiplication region or by thinning the multiplication region down to a few tens of nm. Regarding the research on wide band gap SAM-APDs based on materials suitable for hard X-ray detection, the literature [26,53,54,55] report efficiency values ranging from 20% to 40% in the same energy range, considered in this manuscript. In terms of the dark current, values ranging from 2 to 30 pA can be found for reverse biases from 2 to 10 V, while, in terms of gain, values of *M* up to 22 are reported.

In order to make a comparison with state-of-the-art devices based on other materials, it should be noted that devices using silicon, which exploit the advantage of having an intrinsic value of k≤0.05 [56], are generally not considered suitable for high energies and rather they are increasingly used together with converters in order to achieve high performance in speed. For example, Si APDs operate in optical telecommunication systems by utilizing a Ge absorber (very efficient for wavelengths ≤1550 nm) in a separate absorption, charge, and a multiplication structure that utilises Si as the low-noise multiplication region [57]. Although the gain-bandwidth products of these latest generation devices are very high (up to 340 GHz) [58], they are suitable for very-low-energy photons (near-infrared) and not for hard X-ray detection.

### 3.3. Synchrotron Radiation Measurements

In order to evaluate the role of the thickness of the absorption region and of the metal–semiconductor interface in the charge loss, measurements were carried out with synchrotron light, which allows, by changing the energy, to deliberately tune the depth at which the e-h pairs are generated. For each device, measures were preliminarily performed to chemically characterize them and to confirm what could not be assessed only by an optical inspection, (that there were areas not covered by the anode and that there were no particular defects or impurities that could affect the measurements). Thanks to these initial measurements, it was possible to identify areas that were then used for all subsequent measurements (see marked areas in Figure 7).

For each of the analysed devices, photocurrent maps and X-ray fluorescence maps were acquired at different soft-X-ray photon energies at the TwinMic beamline [59]. Representative images acquired with 2010 eV photons and an optical image are reported in Figure 7.

The top of the mesa is visible in the photocurrent map, where low photocurrents are indicated with dark violet shades and high intensities with light yellow shades, respectively, whereas the central area is obscured by the Al bond wire, which provides the electrical connection of the window electrode to the preamplifiers. The wire is also clearly visible in the simultaneously-acquired X-ray fluorescence map, which displays on a pixel basis the number of measured As L${}_{\alpha }$ fluorescence photons with an energy of 1282 eV (Figure 7b).

To analyse the effect of the Au/Cr–GaAs interface, we measured the currents when the radiation entered the device, passing exclusively either through the surface covered by the contact or directly in the GaAs. The measurements were carried out at a reverse bias of 20 V, far from the appearance of the multiplication (see Figure 5). The ratio between the signal acquired passing through the double metal layer and the one acquired by irradiating directly the GaAs was then compared with the theoretical value of the absorption of Au and Cr calculated with the IMD software [42]. The last two columns of Table 1 report the measured attenuation $T_{m}$ (i.e., $I_{m}^{Au}/I_{m}^{GaAs}$) and the theoretical expected attenuation $T_{th}$. The measured and theoretical attenuation values versus the photon energy have a similar trend, indicating that traps and defects between Au/Cr and GaAs surfaces seem to have small effects in the loss of charge collection efficiency.

The collection efficiency measured for all the devices was lower than the expected value, i.e., the measured current was lower than the one calculated from the flux and the energy of the photons. Since the field in the absorption region was almost negligible, at a first stage, recombination in such a region (where carriers move mainly by diffusion) was considered a major cause of efficiency reduction. In this perspective, as mentioned in Section 2.4.4, the five energies were chosen to have different attenuation lengths, in order to produce most of the carriers at different distances from the multiplication layer (see Figure 3).

Since in both devices with $d_{abs}=4.5\;\mu$m and $d_{abs}=15\;\mu$m, at all considered energies, the photons were almost completely absorbed before reaching the multiplication layer (see Figure 3), the theoretically expected current (in lossless and gainless devices) could be calculated, considering the photon energy $E_{ph}$ and the incident radiation flux $\Phi_{0}$ as
(8)$$I_{th}\left(E_{ph}\right)=\Phi_{0}\;\cdot \;\frac{E_{ph}}{E_{e-h}}\;\cdot \;q,$$
where $E_{e-h}$ and *q* are introduced in Equation (Equation 3). The measured current, irradiating the device directly on the GaAs surface, was compared with the current from Equation (Equation 8) as a function of $\Phi_{0}$ and $E_{ph}$ so that the efficiency η was calculated as $I_{m}^{GaAs}/I_{th}$ (see Table 2).

In Figure 8, we report the change in efficiency as the attenuation length increased. It can be observed that there is no clear dependence on the attenuation length as no evident trend can be observed for both sensors. The modest variations between one energy and another are most likely due to systematic errors. In fact, at each change of energy, it was necessary to reposition the sample; for this reason, each measurement was probably performed on a different position of the device, yielding the observed differences in efficiencies.

Conversely, the differences in the efficiencies between the two types of devices were likely caused by variations in the fabrication process rather than the recombination in the absorption layer, since the calculated efficiencies for devices with $d_{abs}=15\;\mu$m are higher than those of devices with $d_{abs}=4.5\;\mu$m, even if the electron path length before entering the multiplication region is three times larger. Further investigations on several devices of the same type are necessary to obtain more reliable information. It should be noted that neither the absorption region thickness nor the deposition of the anodes appear to have a decisive role in the charge loss, and that the same behaviour is common to all devices having the same thickness. A possible explanation for this behaviour could be ascribed to small uncontrolled fluctuations of the C atom density in the δ p-doped layer from batch to batch. In this regard, it must be pointed out that we are working in a region where δ p-doped C layers are highly compensated [36], and a small excess of deposited C atoms above the optimal dose might increase defect formation due to atomic pairing [60], consequently decreasing the efficiency of the device.

Finally, as we have seen, the measurements made with the laser as well as with the synchrotron light, showed the presence of a photogenerated current at voltages lower than the punch-through voltage; that is, in a condition in which APDs do not commonly show sensitivity to light. This behaviour is a consequence of the deposition of a thin sub-monolayer of C atoms (δ p-doping), and it can be understood by using the same Sentaurus TCAD simulations that reproduce capacitance versus inverse bias: the variations of the potential barrier at the δ layer as the reverse bias increases are displayed in Figure 9. It emerges that, for heavily doped and very thin δ layers (in principle, almost less than a monolayer), the voltage barrier preventing diffusion of electrons from the absorption layer into the multiplication region is very low, and it decreases slowly when the bias is increased. This allows to obtain relatively high photogenerated currents even below the punch-through voltage [35]. In these simulations, the disappearance of the potential barrier and the onset of the punch-through take place at about the same voltages at which the start of breakdown is experimentally observed.

## 4. Conclusions

GaAs-based APDs are valid alternatives to Si-based devices, particularly thanks to their higher carrier mobilities and their good collection efficiencies of high-energy photons. This latter has been extensively investigated. Among the possible causes of decreases in such efficiencies, there are carrier recombinations inside the absorption layers, where charges move only by diffusion (below the punch-through voltage), and carrier recombinations in interfacial regions, where undesired traps may be present. These cases were carefully analysed with the aid of synchrotron radiation, allowing to create the photocurrents at different depths in the absorption layer, and by designing and fabricating devices featuring different thicknesses of the absorption layer. Furthermore, thanks to the availability of a micrometric radiation beam and the presence of entrance windows (not covered by the Cr/Au anode), photocurrents could be generated without interactions between the beam and the electrode; this allowed assessing the net role of the interface between the anode and semiconductor. The acquired data substantially highlight the absence of traps in the interfacial regions and an independence of the efficiency from the thickness of the absorption region. Since the charge collection efficiency does not depend on the latter, we speculate that small fluctuations of the C atom density above the optimal value in the δ p-doped layer changing from batch to batch may generate defects that affect charge loss.

Furthermore, these measurements show that devices engineered not to reach the punch-through (by means of extremely thin delta layers) still exhibit quite high photocurrents, unlike what happens in the canonical SAM-APD configuration, in which the photoinduced current becomes significant only above the punch-through voltage. This last aspect is important when it is necessary to reduce the multiplication noise, since the absence of punch-through prevents multiplication in the absorption region. Furthermore, it must be considered that, at the energies considered here, all the analysed devices have efficiencies greater than what is possible to obtain with Si devices of the same sizes and thicknesses. This, together with the values found for ENF and *M*, which are in line with what can be found in the literature for GaAs-based devices, means that they can detect photons up to a few keV. Finally, our results suggest that it is possible to increase the thickness of the absorption region without severe drawbacks and to work without the depletion of the absorption region.

## Figures and Tables

**Figure 1 sensors-22-04598-f001:**
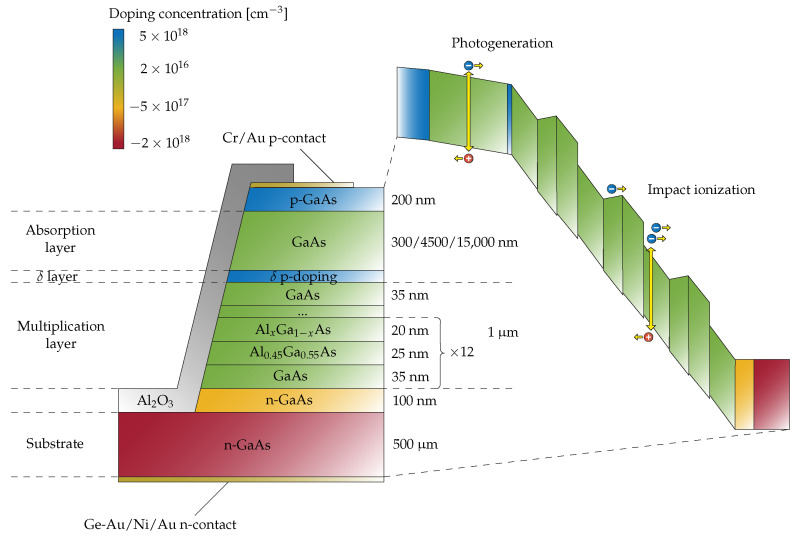
Sketch of the GaAs APD considered in this work (not to scale). The grown layered structure is depicted on the left side, where the layer colour represents its doping concentration. On the right side, there is a corresponding band diagram under the reverse bias.

**Figure 2 sensors-22-04598-f002:**
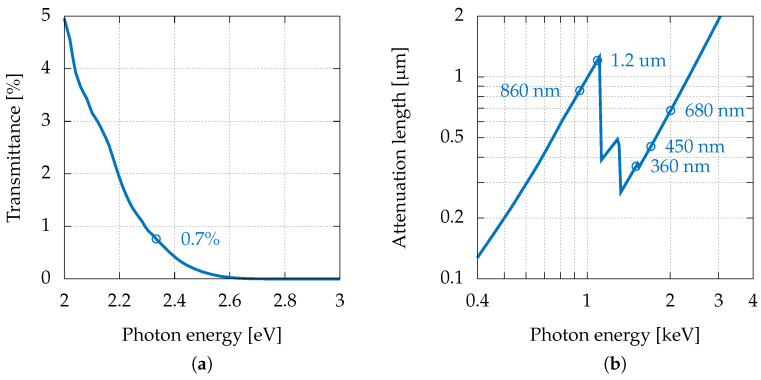
(**a**) Transmittance of 500 nm of GaAs versus photon energy from 2 to 3 eV obtained with IMD software [42]. (**b**) Attenuation length in GaAs versus X-ray energy from 400 eV to 4 keV [43]. The marked points indicate the energies at which the measurements were performed.

**Figure 3 sensors-22-04598-f003:**
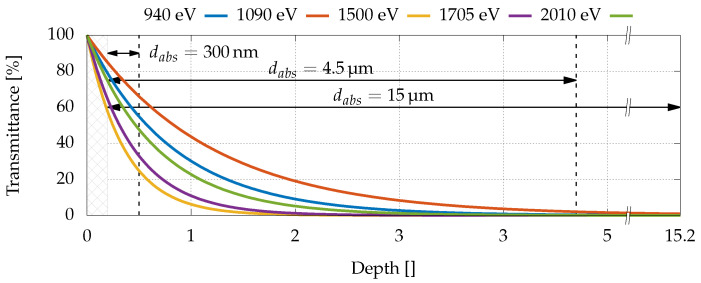
Transmittance as a function of the absorption layer width at different radiation energies. The 200 nm p-GaAs layer is grayed out.

**Figure 4 sensors-22-04598-f004:**
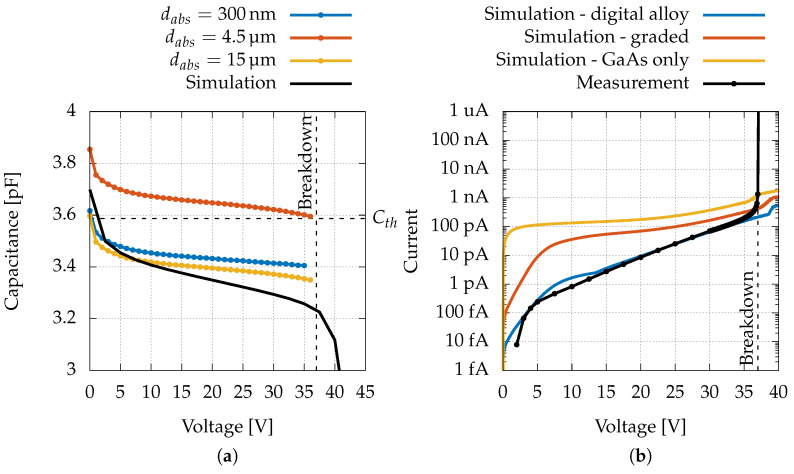
(**a**) C-V simulation obtained with Sentaurus TCAD software suite, C-V measurement of devices with different absorption region thicknesses and theoretical capacitance ($C_{th}=3.59\;pF$) of a a parallel-plate capacitor. (**b**) Dark I–V curves obtained with different models of the multiplication region.

**Figure 5 sensors-22-04598-f005:**
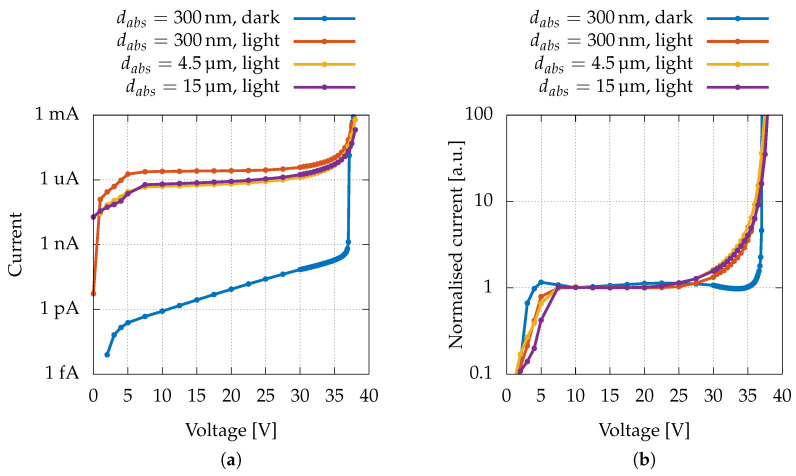
(**a**) Dark I–V characteristics of the device with $d_{abs}=300\;$nm compared to light I–V measurements ($P_{laser}\approx$ 50 μW, λ = 532 nm) of the three types of devices. Dark currents of the other types of devices present no appreciable differences with the one presented in this graph on this scale and were omitted for clarity. (**b**) Normalised I–V measurements.

**Figure 6 sensors-22-04598-f006:**
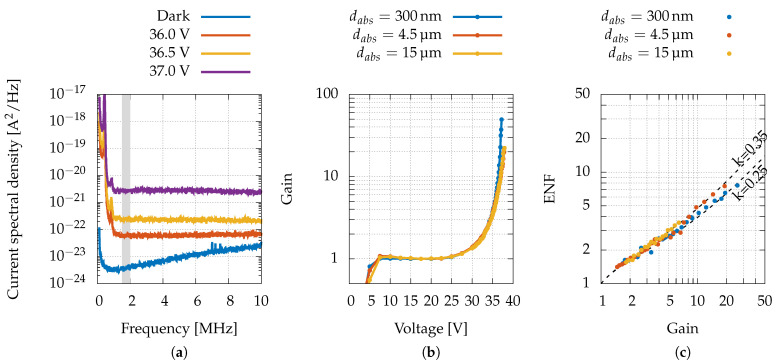
(**a**) Noise spectra acquired at different bias voltages with a fixed laser intensity. The grey-shaded band shows the frequency range used for ENF estimation. (**b**) Gain as a function of the polarisation voltage for the three device types. (**c**) Calculated ENF for the three device types compared with the theoretical trend of the local model [47] with k=0.25 and k=0.35.

**Figure 7 sensors-22-04598-f007:**
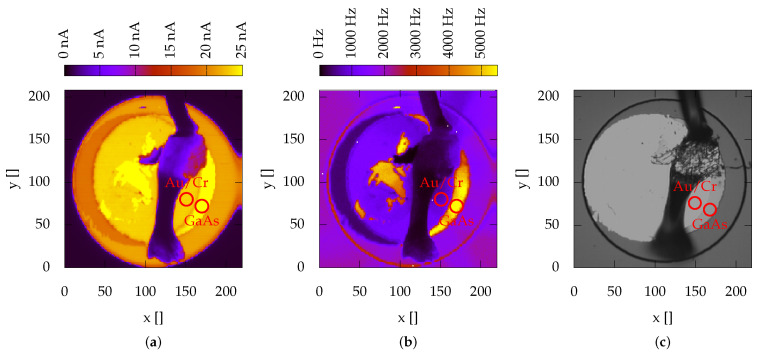
(**a**) Photocurrent map; (**b**) X-ray fluorescence image of the As L${}_{\alpha }$ line; (**c**) microscope optical image of the device. Regions of higher intensities correspond to the unexposed half-moon shaped areas of the devices, which are clearly visible in all three images.

**Figure 8 sensors-22-04598-f008:**
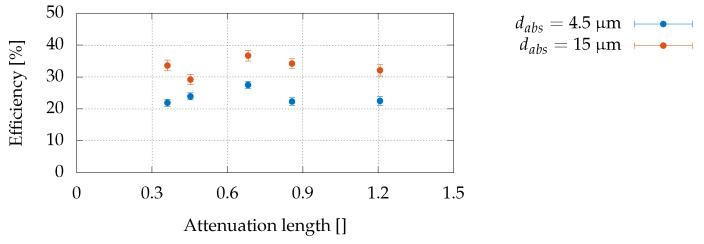
Efficiency as a function of attenuation length.

**Figure 9 sensors-22-04598-f009:**
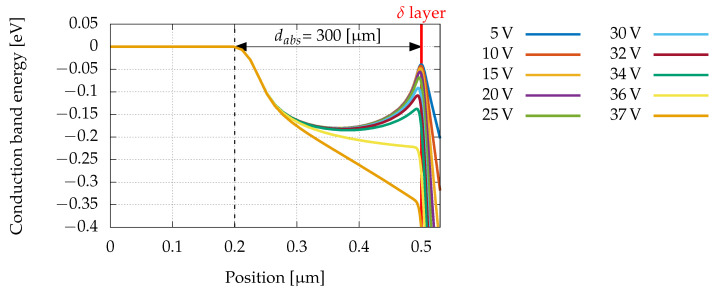
Simulated conduction band profile for different reverse biases.

**Table 1 sensors-22-04598-t001:** Measured currents at different photon energies with devices with $d_{abs}=4.5\;\mu$m. For each energy, we measured the current illuminating the device through the Au/Cr contact (column 2) and directly in the GaAs layer (column 3). From these values, we calculated the transmission of the Au/Cr contact (column 4) and we compared it with the theoretical value obtained with IMD software (column 5). Of note is that the fluctuations associated with the photon energy were governed by systematic errors, which were in the order of 5 eV.

Energy [eV]	$I_{m}^{\mathrm{Au}}$ [nA]	$I_{m}^{\mathrm{GaAs}}$ [nA]	$T_{m}$ [%]	$T_{\mathrm{th}}$ [%]
940	64.187±0.025	119.793±0.025	53.58±0.02	51.8±0.3
1090	81.903±0.025	125.670±0.025	65.17±0.02	62.0±0.3
1500	232.192±0.025	318.681±0.025	72.86±0.01	79.3±0.2
1705	293.290±0.025	378.027±0.025	77.58±0.01	84.6±0.1
2010	278.579±0.025	322.070±0.025	86.50±0.01	89.1±0.1

**Table 2 sensors-22-04598-t002:** Measured currents at different photon energies. The expected theoretical currents (column 5) were calculated from the photon energy and the photon flux by using Equation (Equation 8). The ratio between the measured and theoretical currents is reported in column 6. As before, the fluctuations associated with the photon energy were governed by systematic errors, in the order of 5 eV.

$d_{\mathrm{abs}}$ [μm]	Energy [eV]	Flux [$10^{10}$ Photons/s]	$I_{m}^{\mathrm{GaAs}}$ [nA]	$I_{\mathrm{th}}$ [nA]	$\eta =\frac{I_{m}^{\mathrm{GaAs}}}{I_{\mathrm{th}}}$ [%]
	940	1.50	119.793±0.025	537±27	22.3±1.1
	1090	1.35	125.670±0.025	561±28	22.4±1.1
4.5	1500	2.55	318.681±0.025	1457±73	21.9±1.1
	1705	2.43	378.027±0.025	1578±79	24.0±1.2
	2010	1.53	322.070±0.025	1172±59	27.5±1.4
	940	1.60	195.530±0.025	573±29	34.1±1.7
	1090	1.95	259.754±0.025	810±40	32.1±1.6
15	1500	3.30	634.934±0.025	1890±94	33.7±1.7
	1705	2.37	449.844±0.025	1539±77	29.2±1.5
	2010	1.53	430.337±0.025	1172±59	36.7±1.8

## Data Availability

The data are available upon request.

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
