# Peer review of "Synchrotron Radiation Study of Gain, Noise, and Collection Efficiency of GaAs SAM-APDs with Staircase Structure"

_sensors, 2022, doi:10.3390/s22124598_

Round 1

Reviewer 1 Report

The authors report on the characterization of GaAs based avalanche photodiodes using laser and synchrotron radiation. Some of the results are compared to simulations.

Despite I do believe that a lot of work was put into the measurements, the data analysis as well as simulation I unfortunately have to say that in my opinion much less effort has been put into the concise writing of the manuscript, which not only lacks an overall aim but also widely comprehensable content. Given such a large number of (co-) authors a more rigid internal review process should have taken place by more experienced colleagues before releasing the manuscript for submission. Maybe some specialists in the field can have a deeper understanding what the authors are presenting, but as soon as a broader audience is concerned I believe many questions will remain open.
Thus I unfortunately have to suggest the paper to be published only after major revisions.

Find my detailed questions/comments below, referring to line or Figure/table numbers of the draft manuscript.

General:
In my opinion 19 pages are much too much for a research paper which could/should be cut by half, focussing only on the essential points. Despite a lot of work and characterization has been done, not all of it is neccessarily required to be published/presented to the audience to (in such detail) 
--> consider shortening, especially you should be critically asking which parts are really needed for the reader to get the essencial results of your work; maybe also leave out some parts for which you obviously have difficulties explaining the observed results.
Sometimes you also have to be more specific (e.g. give numbers when you talk about some parameters etc.) and leave out "colloquial" formulations
The general manuscript and also the language (maybe coming from translation programs) are sometimes quite imprecise and in my opinion almost do not fulfill the required scientific soundness! 
Basic principles of writing scientific publications like coherent and comprehensible explanations, precise formulations and proper critical discussions and comparison/ranking of the results are partially missing! Also consider reviewing the langue or have a native speaker proof read the manuscript.

The abstract doesn't really match the introduction nor what is presented in the manuscript! Nothing about the X-ray applications can be found in the latter (especially not for energies above 2 keV!), and it is also not specified throughout the manuscript what your devices should be applied to/what you are developing them for. I think this is crucial for the understanding! 
It is further not clear to me at all what the overall aim of this manuscript or its novelty is; if there are interesting or promising features they are not "sold" well enough to the audience.

line 1: which applications?
5: which cases?
13: which capabilities in which operations?
14 what are the promising performances? this is nowhere discussed later!

19-21: far too little references given.
27: prerogative maybe the wrong word here?
32-36: isn't this sentence self-contradictory concerning the wavelengths? Consider rephrasing.
41: would be nice to give the numbers of these properties, or even compare them to Si.
45 ff: if you write about GaAs and X-ray applications you should also refer to the many promising results from the recent years using other GaAs technologies (Cr-compensated GaAs, photon counting and charge integrating readout electronics etc.).
49: "endeavours have been undertaken"--> citations available?
69 ff: again: what is the purose of your devices? What are you aiming to do with them?
76: where is there a comparison to past literature? I didn't find that at all!
Eq. 1: isn't the E-power dependency of the photoelectric effect rather 3 or 3.5 than 3/2?
94: better give absolute values of lambda and not "20 times shorter", especially since you have a very thin absorption layer in your structures compared to other X-ray detectors.
95: for even higher field the velocity even decreases again as far as I know, so there should be a certain field strenght for v_max (few kV/cm or so)!
104/109: your layers are only max. 15 µm thick if Fig 1 is correct, so you will have less than 30 % absorption in this layer and non-negligible absorption in the multiplication layer for higher X-rays! -- comment!
115: what do you mean by "is worth" and "is kept low"? Be more specific.
131-133: What were the results of these studies and how did they influence your work/design? The sentence just as you wrote it doesn't help the reader at all.
135: what were these thicknesses? Be more specific here.
152: what do you mean by "is placed"? In a controlled way by you defining the p-doping concentration?
155 : what do you mean by "somewhat different"? In what sense? As compared to what? What are you talking about then in lines 141-154? I thought this is what you have done as is indicated in line 141-144? If there is ay difference to anything else you need to specify! This part is quite confusing to me, also concerning which parts are/should be depleted and which not at which voltages in which devices. Please rephrase and try to avoid words like "somewhat" in scientific manuscripts!
164: where did yo discuss that previously? In line 96 there are some 100 µm mentioned which is far away from the thickness here! Be more precise. You also have to comment on the fact that with decreasing absorption layer thickness you have increased absorption in your multiplication layer with all its consequences!
167: avoid expressions like "somewhat" in scientific manuscripts!
168 what is the consequence of this? In one part you then have additional ansorption in the gold, in the other part not (compare Eq 2). If this can be neglected, why do you mention it at all? Or why don't you cover the full surface then with the contact? Was this on purpose or an accident in the deposition?
189: what do you mean by "closely match"? I can see quite a difference between the lines! What was the error in the experiments?  What could explain this difference, especially at higher voltages >15 V? Is this the "range" that you speak of in line 193? --> Be more precise and consider rephrasing lines 189-194, or show what you mean in the Figure e.g. by arrows, circles or so! 
--> by the way, you are still in the Materials and Methods section here. There should be no results shown yet (especially since you describe your actual measurements only in the next paragraph!)! --> move to section 3
195: what do you mean by "upard curvature"? Which of the simulations show this trend in your opinion? There are three different ones in Fig 2b. Be more precise or mark it in the Figure e.g. by a circle!
197: introduce the SRH abbreviation in line 186.
211ff: what is the accuracy/error of your measurement setup? If applicable, add error bars to your experimental plots (throughout the manuscript) or discuss the errors.
Fig3: is it really required to have a Figure almost one page large to explain the readout electronics? What is the added value to the reader? As mentioned in the beginning, the paper is way too long in my opinion, and this could be one way to shorten it.
235: what do you mean by "some eV"? According to Fig 4a you had a laser with one specific energy used for your measurements? Why don't you specify/describe this here in more detail?
254ff: what is the added value to the reader from the fluorescence measurements? Given the length of the paper this part could/should be completely omitted.
260: how precisely do you know this photon flux? Is it really with higher precision than the one from the laser (line 244)?
261ff : obviuosly there is an absorption edge of GaAs in the energy range you employed, which means you generate fluorescence photons when using the three higher energies, which themselves potentially have considerably different attenuation lenghts! What is their impact on your measurements or your conclusions? Also, you are talking about high energy X-ray applications in your introduction. Why didn't you do any measurements at such energies around 20 keV or so? How realistic/transferable are your conclusions from your measurements there?
Fig 5: what do you conclude from these calculations? Also, do not use abbreviations like "Ch. density" on the axis. Better use 2 lines or smaller fonts and write "Charge densities" or simply lambda.

Results section:
general: in my opinion the whole result and discussion section is insufficient for publication and has to be reworked, especially the synchrotron measurement part! You present some results without profound discussions or conclusions or explaining what the reader should learn from that; quite often you just give a rough estimate or guess what could cause your observations, which is not very scientific and should be clarified before publishing them!

272: the recorded value at zero voltage?
276-278: as said before, these results should be placed and discussed here. Maybe you can even combine Fig 2 and Fig 6 into one Figure
Figure 6 caption and line 283: "The capacitances of all devices are essentially the same"? As long as errors are not discussed that may be true for the 300 nm and 15 µm structure, but obviously the 4.5 µm device has considerably higher capacitance (so even there is no trend with thickness?)--> comment!
281/282: would the depletion of the absorption layer be something that you want or not? (see also comment above regarding line 155)
Fig 7: why don't you show the dark measurements for all thicknesses?
303/304: in this case you could do some statistics and add error bars to your plots.
342-353: so what is the conclusion from all this? This would be quite important for the reader...
355-371: what do we learn from all that? Is that essential for the understanding?
line 356 and 380: no need to repeat the energies since you gave them already in the section before.
367/368 and higher fluorescence signal
372: which effect are you looking for? As you stated above, for higher energies it can be neglected. See also comment regarding line 168

383: "within the uncertainties" --> this is not true! you give errors of 0.01 and 0.02 percent whereas the differences are somewhere between 2 and 5 percent (a actor of 100!)! maybe a similar trend can be observed...
384/385: what do you mean by that? Do you observe any of that?
386: which efficiency reduction are you talking about? Are you observing that? Where is that discussed?
table 2 caption: why do you discuss the uncertainty of the photon flux here and not in the methods section? How is that value as comparesd to the laser? 
387: should the charge carriers in the absorption layer move only by diffusion? In normal X-ray detectors charge carriers move by external fields? I'm confused here.
401ff : could the excitation of fluorescence photons for the higher energies play a role? Or the difference in the line profiles in Fig 5 for the two lower energies as compared to the higher energies?
409 ff: what are the variations in fabrication process? I think given the fact that the distance is 3 times larer and still the efficiency is higher you should provide at least a more elaborated guess or theory what could cause this behaviour!
414ff: it is not clear to me why you characterise this thin device at all, if even for such low X-ray energies you already have problems with it.
430-440: even after having read this paragraph multiple times I do not get at all what you are saying here. Please rewrite much more comprehensible and discuss the results of the 300 nm device vs the thicker devices more profoundly

464: is it really a good efficiency? Are 2 keV really high energy photons?
465ff: I disagree! You give many possible reasons or guesses but no real scientific proof or any profound discussion (e.g. comparison to the efficiency Si-based devices or other devices from literature!). You give two possible reasons for reduction on efficiency (recombination and interface) but you showed that both are basically negligible --> Why then do you have such a reduced efficiency? This needs to be discussed!
481: anyhow??? In't that your job to explain why this is the case?

You mention noise, gain and efficiency investigations in your abstract and introduction, but do not mention at least noise and gain results (and their disussion or comparison) at all here.

What is the real conclusion of all this work/manuscript? What are the devices useful for? Can they be applied to science cases or are they purely prototypes in development? What needs to be adressed/improved/understood better in the future? How do your devices perform compared to other devices (other materials or other structures presented in literature)?
If the real novelty of all this is the use of the thin, highly doped delta layer and the operation in non-punch-through regime, this should be sold to the audience much better! Or maybe this is even causing some of the effects that you see but cannot explain??

Reviewer 2 Report

The authors presented the theoretical study and experimental characterization of three series of GaAs APDs with varied absorption regions of different thicknesses. The simulation and experimental sections are clear, however, a few comments as followed should be addressed before being published.

  1. To reduce confusion for readers, some abbreviations may not be necessary, such as SPDs, SiPMs, and SDD, which are only appeared once or twice.
  2. The introduction should have more discussion on recent work on III-V-based APDs, especially GaAs-based APDs, and pointed out the novel of this work compared to others.

Reviewer 3 Report

The manuscript presents a analysis of the radiation detector performance, noise and the efficiency considering critical design points of GaAs avalanche photodiodes (APD).  The authors performed the design, the fabrication of 3 kinds of APDs and evaluation of the device performance for laser and X-ray photon detection. The experimental characteristics were compared with TCAD simulation results. The manuscript is well-organized and written with clear figures and explanations. The results are important for advanced applications of GaAs-based APDs. I would recommend the manuscript for publishing. Though, there are few details which need to be corrected for the sake of clarity.

Comments

The title, the meaning of “SAM-APDs” is unclear. This abbreviation is a specialized term, and it may have many interpretations. There is no explanation of this new term the abstract too. Please reconsider.

In the text you are showing the performance for your devices only. How does it compare with the-state-of-the-art Si-based devices? Please add.   

Eq.1 and Eq. 2 need references.

Section 2.2 what is the growth conditions - temperature, gas pressure? Adding a reference will be useful.

Section 2.3 what is your mesh size, does it uniform? Give some details on the simulation model structure.

There are few English flaws. An English proofreading is advisable.

Round 2

Reviewer 1 Report

Despite I think that some parts still lack precision or make it difficult for the reader (especially those not being experts in the field of SAM-APDs) to follow the authors' intention, the manuscript clearly has improved in its revised version.
Most concerns/comments/questions of the reviewer(s) have been included, so it leaves only a few open points for me which I would like to be addresed before I can recommend the paper to be published.

general comments:
- many Figures (and also tables) have missing units in the brackets now (probably some font issue) --> carefully check all units in the manuscript!
- still try to avoid excessive use of expressions like "in fact" etc.
- still I don't understand why you bring up hard X-ray applications at all and don't restrict yourself to the few keV that you use throughout the manuscript (hard X-rays are also very rarely referred to in the later parts of the manuscript). I also believe that for much thicker absorption regions needed for acceptable efficiencies for energies of the order of ~20 keV you would find many other effects much different from what you see now, that you will not be able to explain by simply "extrapolating"your findings here (like much longer diffusion lengths and thus possible recombination in the absence of electric fields, problems caused by long-ranged k-edge fluorescence photons for photon energies above ~12 keV etc)
- still in my opinion the (quantitative) advantage of your design (no punch-through) as compared to state-of-the art devices is not really clear (how much better is the gain/noise/efficiency?), and also regarding the high-speed/time-resolved applications (diffusion of electrons versus bias-driven drift), which you do not investigate or discuss at all
- the possible role of the fluorescence photons that are present for the higher energies is not discussed
- the gain itself (despite being "promised" in the introduction) is not really mentioned quantitatively (only mentioned in the ENF section and in Fig 5b the reader has to extract it by him/herself)

specific comments:

line 29 and line 72: be very careful not to mix up single photon detectors (i.e. single photon sensitivity) and single photon counting detectors! This is something completely else and should not be confused or even compared!
line 100: "good collection efficiencies" --> do you really have them? what is the quantitative comparison to other devices? "without full depletion"--> I thought you measure at conditions where no depletion at all is present?
line 156 p-doped "delta" layer?
line 169: I'm confused here with "but still ensuring complete depletion of the absorption layer" --> I thought from what is written directly above (line 163/164) that the depletion of the absorption layer only starts at applied bias voltages larger than the punch-throug voltage??? Don't you operate at voltages sufficiently below punch-through? maybe the properties and specific working conditions of your deveices versus "standard" SAM-PADs should be made clearer.
Figure 6: the meaning of "McIntyre's theoretical trend" is not introduced in the text; why are there no higher gain values >6 for the thickest structure and no lower gain values <4 for the other structures shown? why does the k-value appear to suddenly change for the red curve at M ~7 whereas for the other devices it seems to be either 0.25 or 0.35?
line 397: which predefined biases? how and why did you choose these settings?
406-410: again in my opinion you should restrict the paper to these energies for the reasons mentioned abve
415 ff: Ge has the same "effective Z" value as GaAs. Why should it not be suitable for (hard) X-ray applcations then if they had the same thickness?
420ff: what do you want to say with that? what is the conclusion of that?
422: bandgap instead of gap?
425 ff: what do you want to say with that? what is the conclusion of that? are these devices better/worde/different than your devices and if yes, in which aspect?
440/441: it would be nice if you could mark these regions in the Figure 7 since obviously in the central area covered by the contact you have large variations in current as well as fluorescence signal. Could these be mechanical damages from the wirebonding? How can these regions effect your measurements (e.g. effecive variations of the thickness or even damage of the contact in these areas etc.)? how precisely do you know the contat thicknes (which obviously would have a large influence on the theoretical T_th values)?
482-484: see previous comment. Also, as mentioned before, a comment on the possible or negligible influence of the generated fluorescence photons which are not present for the two lower energies would be nice.
489 ff: in my opinion a more quantitative discussion would be nice than rather only explaining the reason for potential efficiency osses: are these ~20-35 % efficiency still good or very bad or something very unexpected or how do these values compare to the other more common devices you mention above?
497/498: the logical step between these paragraphs is quite big and may confuse the reader. Arew you still talking about the synchrotron measurements here or which part is this related to? the dark/light measurements from Fig 5 were done with the laser... consider putting this paragraph somewehere else (or omitting it since it is not clear to me what we learn form it)
504: how low? how does that compare to other, more common devices?
506-508: "In these simulations, ..... is experimentally observed." Isn't that kind of self-contradictory?
512/513: it's a bit confusing to first talk about good collection efficiency and then causes for decreased efficiency (see also comment above)
527: what do you mean by "changing from sample to sample"? in this case you would have much different results between different samples which I had the impression was not the case?
530: how high? especially as compared to the "canonical SAM-PAD configuration" --> as mentioned above, it would maybe be better to explain a bit better in the introduction the difference between your devoces and the "common" SAM-PADs (e.g. in terms of thickness of the delta layer, working conditions and field distribtutions under these conditions etc.) This would largely help the reader to understand the significance of your results!
539-541 I'm not so sure about that when it comes to thicknesses required for efficient absorptions of higher energy photons of ~20 keV
